# Anti-ZSCAN1 Autoantibodies Are a Feasible Diagnostic Marker for ROHHAD Syndrome Not Associated with a Tumor

**DOI:** 10.3390/ijms25031794

**Published:** 2024-02-01

**Authors:** Akari Nakamura-Utsunomiya, Kei Yamaguchi, Naoki Goshima

**Affiliations:** 1Department of Pediatrics/Medical Genetics, Graduate School of Biomedical and Health Sciences, Hiroshima University, Hiroshima 734-8551, Japan; 2Department of Pediatrics, Hiroshima Medical Center, Asa Citizens Hospital, Hiroshima 731-0293, Japan; 3Division of Neonatal Screening, National Center for Child Health and Development, Tokyo 157-8535, Japan; 4Molecular Profiling Research Center for Drug Discovery, National Institute of Advanced Industrial Science and Technology, Tokyo 135-0064, Japan; 5ProteoBridge Corporation, Tokyo 135-0064, Japan

**Keywords:** zinc finger and SCAN domain-containing protein 1, subfornical organ, Na_x_ channel, rapid-onset obesity with hypoventilation, hypothalamic and autonomic dysregulation syndrome, autoimmunity

## Abstract

Recent studies have reported the presence of autoantibodies against zinc finger and SCAN domain-containing protein 1 (ZSCAN1) in the sera of patients with rapid-onset obesity with hypoventilation, hypothalamic and autonomic dysregulation (ROHHAD) syndrome associated with neuroendocrine tumors, suggesting immunologic and paraneoplastic processes as the pathologic underpinnings. Moreover, several hypothalamic regions, including the subfornical organ (SFO), were reported to exhibit antibody reactivity in a patient with ROHHAD syndrome not associated with a tumor. Whether ROHHAD syndrome not associated with a tumor is associated with anti-ZSCAN1 autoantibodies remains unclear. We used a comprehensive protein array analysis to identify candidate molecules in the sera of patients with ROHHAD syndrome and identified ZSCAN1 as a target antigen. We also found that ZSCAN1 was co-expressed at the site of antibody reactivity to the IgG in the patient serum observed in mouse SFOs and an enzyme-linked immunosorbent assay showed that >85% of the patients with ROHHAD syndrome were positive for anti-ZSCAN1 autoantibodies. These results suggest anti-ZSCAN1 autoantibodies as a feasible diagnostic marker in ROHHAD syndrome regardless of the presence of a tumor.

## 1. Introduction

Rapid-onset obesity with hypoventilation, hypothalamic and autonomic dysregulation (ROHHAD) syndrome, initially described by Ize-Ludlow et al. in 2007 [1], typically presents with rapid-onset obesity starting at an average of 3 years of age, hypophysial involvement (growth hormone deficiency, central hypothyroidism, and hypothalamic involvement, including central apnea and attention-deficit hyperactivity disorder), and autonomic nervous system symptoms [2,3]. Although rare, ROHHAD syndrome is associated with an increased risk of sudden death [4]. Recent studies have proposed several potential etiologies including genetic events, such as imprinting abnormalities, and autoimmunity [5,6,7,8]. Moreover, autoimmunity to the hypothalamus and periventricular organs, including the subfornical organ (SFO), has been reported as an underlying pathologic mechanism [9]. In 2022, Mandel-Brehm et al. reported the presence of antibodies against zinc finger and SCAN domain-containing 1 (ZSCAN1) in the serum and tumor samples of a patient with a neuroendocrine tumor (NET) and ROHHAD syndrome (ROHHAD-NET) [10]. Moreover, one study demonstrated that antibodies against the Na_x_ channel, which senses sodium levels in body fluids, were related to the pathophysiology of ganglioneuroma in a patient with ROHHAD-NET [11]. On the other hand, the relationship between ROHHAD syndrome in patients without tumors and the presence of anti-ZSCAN1 autoantibodies has not been elucidated.

In the present study, we aimed to perform the detection of autoantigens by means of a protein array analysis in patients diagnosed as having ROHHAD syndrome not associated with an NET. Moreover, we hypothesized that anti-autoantigen reactivity identified via enzyme-linked immunosorbent assays (ELISAs) could be a diagnostic marker in this syndrome without tumors.

## 2. Results

### 2.1. Protein Array Results

A mixture of serum samples from Cases 1–3 and a mixture of serum samples from Cases 2–4 (Table 1) were analyzed to identify potential autoantigens using the comprehensive protein array. After the exclusion of proteins to which the serum samples of both the patients and the normal controls exhibited antibody reactivity, 33 proteins were identified as candidate antigens. A custom protein array including these 33 proteins was used to analyze the serum samples of all five patients with ROHHAD syndrome (Case 1–5). As shown in Table 2, ZSCAN1 was identified as an antigen in four of the patients with ROHHAD syndrome not associated with a tumor and was not an antigen in the only patient (Case 4) who was diagnosed with ROHHAD syndrome associated with a tumor.

### 2.2. Immunostaining of the Mouse SFO

Subsequently, we analyzed the expression of patient IgG and anti-ZSCAN1 in mouse SFO. The serum sample from Case 1 and an anti-ZSCAN1 antibody were used for the immunostaining of the mouse SFO. As shown in Figure 1A,B, the IgG in the patient serum sample showed a positive response to the mouse SFO, which is in agreement with previous reports. In addition, ZSCAN1 was expressed in the mouse SFO. The observed co-reactivity to both the patient serum IgG and the anti-ZSCAN1 antibody in the same portion of the mouse SFO suggested that the IgG in the patient’s serum sample exhibited reactivity to ZSCAN1 in the SFO.

### 2.3. ELISA

Finally, we analyzed the anti-ZSCAN1 autoantibody positivity in a cohort of 14 patients with ROHHAD syndrome not associated with a tumor, 15 normal subjects, and 5 subjects with autoimmune disorders (Table 3). Using our novel ELISA, an index over 40 was considered a positive response. In this cohort, 12 of the 14 patients (85.7%) with ROHHAD syndrome were positive for anti-ZSCAN1 autoantibodies (Figure 2). The average anti-ZSCAN1 autoantibody titer was 115 ± 85. Notably, the patients with a poor prognosis had higher than average titers. On the other hand, the normal subjects and those with autoimmune disorders were negative for anti-ZSCAN1 autoantibodies.

## 3. Discussion

In the present study, we found that the serum of the patients harbored antibody reactivity to the SFOs of mice, which did not exhibit any structural changes in magnetic resonance imaging using protein array analysis. We identified ZSCAN1 as an autoantigen in patients with ROHHAD syndrome without tumors via protein array analysis and demonstrated that ZSCAN1 was co-expressed in the SFOs in mice by means of immunostaining using patient serum samples. Using a quantitative custom-made ELISA for the detection of anti-ZSCAN1 autoantibodies, we found that over 85% of the patients with ROHHAD syndrome without an NET exhibited a positive response to the anti-ZSCAN1 autoantibodies compared with that of the normal subjects. These results suggest that the anti-ZSCAN1 autoantibody level is a feasible diagnostic marker in patients with ROHHAD syndrome not accompanied by a tumor.

The diagnosis of ROHHAD syndrome is primarily based on clinical symptoms including rapid-onset obesity with hypoventilation, hypothalamic involvement, and autonomic dysregulation [1]. Therefore, many patients are diagnosed after the progression of symptoms and some patients have poor prognosis, such as respiratory failure or sudden death [1,3,4]. Therefore, diagnostic methods other than clinical evaluation are solely needed to improve the prognosis in patients with ROHHAD syndrome.

Since 2010, several studies have reported cases of ROHHAD syndrome as a paraneoplastic syndrome associated with autoantibodies in patients with tumor complications [9,10,11]. In the first reported patients with ROHHAD syndrome and anti-Na_x_ autoantibodies, the pathologic lesion was in the SFO [11]. The SFO has various neural networks that connect to the hypothalamus and pituitary gland, including the thirst center, and plays a role as a sensor that detects molecules such as sodium in blood vessels [12,13,14]. Similarly, Mandel-Brehm et al. reported that patients with ROHHAD-NET syndrome were positive for anti-ZSCAN1 autoantibodies [10] and that anti-ZSCAN1 autoantibodies were mainly observed in patients with tumors such as ganglioneuroblastoma, ganglioneuroma, and neuroblastoma, although one patient was positive for autoantibodies even in the absence of a tumor. In contrast to a previous study, our analyses suggested that anti-ZSCAN1 autoantibodies were also present in patients with ROHHAD syndrome not associated with a tumor. Interestingly, we previously reported that the serum of these patients exhibited antibody reactivity against the SFO. The analysis using Western blotting could not identify the target molecule recognized by the patient serum; we hypothesized that either the amount of the molecule was low or the molecule was easily experimentally inactivated. Therefore, we performed a comprehensive protein array analysis to identify candidate molecules in the sera of the patients with ROHHAD syndrome not associated with a tumor. After determining ZSCAN1 as a candidate target, we developed an ELISA with immobilized ZSCAN1 to easily detect antibody reactivity in a large cohort of patient samples. Our analyses indicate anti-ZSCAN1 autoantibodies as a marker for early diagnosis that may also be useful in evaluating disease severity and the need for immunosuppressive treatment in patients with ROHHAD syndrome not associated with a tumor.

The underlying causes of anti-ZSCAN1 autoantibody production in patients with ROHHAD syndrome not associated with a tumor are unclear. Studies have reported patients exhibiting ROHHAD syndrome-like symptoms after acquiring coronavirus disease 2019, suggesting immune activation associated with prior infection as a potential cause [15]. Although the presence of autoantibody production has not been investigated in reported cases, the disease may develop in conjunction with a prior infection. We previously reported patients who were diagnosed with ROHHAD syndrome after influenza or varicella infection. The relationship between ROHHAD syndrome and drugs or vaccinations in unclear, although prior viral infection was recognized as a trigger for inflammation in these patients [11].

In the present study, 12 of the 14 patients were positive for anti-ZSCAN1 autoantibodies; however, the underlying pathology was unclear in the remaining two patients with ROHHAD syndrome who were negative for anti-ZSCAN1 autoantibodies. Our protein array identified additional antigens other than ZSCAN1; therefore, it is possible that antibody reactions to other antigens might influence our results. In our samples for the protein array, one case did not display the presence of anti-ZSCAN1 antibodies, in spite of a lymphangioma association. We speculate that this tumor type was not derived from neuronal cells. Other than autoimmune factors, congenital central hypoventilation syndrome presenting with the PHOX2B variant can be considered in differential diagnoses [16,17]. Furthermore, although the number of reported cases is small, some studies have reported differences in the symptoms of ROHHAD syndrome in monozygotic twins, and the role of imprinting abnormalities should be investigated in future studies [18,19,20].

## 4. Materials and Methods

### 4.1. Subjects

Serum samples from five pediatric patients, including four girls and one boy, were analyzed via the comprehensive protein array (Table 1). Case 4 was complicated by a retroperitoneal tumor. All patients exhibited a specific antibody response to the SFO but were negative for anti-Na_x_ autoantibodies. In addition, serum samples from 14 patients diagnosed with ROHHAD syndrome without tumors [12] and 15 control subjects including 5 patients with other immunological diseases were evaluated using ELISAs for anti-ZSCAN1 and anti-Na_x_ autoantibodies. Furthermore, five additional subjects diagnosed with autoimmune disorders, including two patients with Basedow’s disease, one patient with insulin-dependent diabetes mellitus, one patient with myasthenia gravis, and one patient with paraneoplastic syndrome, were analyzed using our custom ELISAs. Detailed clinical data on patients with ROHHAD are presented in Table 2.

### 4.2. Protein Arrays

In the present study, the HuPEX^TM^ human protein array was used for the comprehensive analysis of autoantibodies in human serum samples. This array contains approximately 13,000 proteins on two glutathione-bonded glass substrates (78 mm × 120 mm). The N-terminal glutathione S-transferase (GST) fusion protein was loaded in two spots, each in a non-dried state that maintained its native structure without drying. The proteins in the array were synthesized from a proteome-wide human cDNA library (HuPEX^TM^) using a wheat germ cell-free system.

For each array, 45 µL of serum from three samples was mixed, diluted to 20 mL, and incubated with the array for 1 h at room temperature. After washing the array with Tris-buffered saline with Tween 20 (TBS-T), the presence of autoantibodies was detected by incubating the array with an Alexa Fluor 647-conjugated goat anti-human IgG (H+L) cross-adsorbed secondary antibody (1:1000 dilution; catalog no:A-21445 ThermoFisher Scientific, Waltham, MA, USA) for 1 h at room temperature. After washing with TBS-T and distilled water, the array was dried, and fluorescence signals were captured using a fluorescence scanner (FV-1000D; Olympus, Tokyo, Japan).

Additional analyses were conducted using a customized protein array containing 33 antigens identified through the comprehensive protein array analysis, a positive control (human IgG), and a negative control (water instead of mRNA). The serum reaction solution included the serum sample (10.5 μL), the diluent (3.5 mL), and skim milk (0.1%). The custom protein arrays were incubated with the serum reaction solutions for 1 h at room temperature. Next, the arrays were washed three times with TBS-T and incubated with goat anti-human IgG (H+L) Cross-Adsorbed Secondary Antibody, the Alexa Fluor 647-conjugated goat anti-human IgG (H+L) secondary antibody (1:1000 dilution; catalog no:A-21445 ThermoFisher Scientific) for 1 h at room temperature. Then, the arrays were washed twice with TBS-T, washed again with distilled water, and dried. Finally, fluorescent signals were detected using a fluorescence scanner.

### 4.3. Immunostaining of Mouse SFO

All animal experiments with animals were performed according to the guidelines of Hiroshima University (no: A17-157). Antibody response in mouse SFOs was evaluated by means of immunohistochemistry as previously described [8,10]. Briefly, 10–16-week-old wild-type male mice (C57Bl/6J; CLEA Japan, Tokyo, Japan) under deep anesthesia were transcardially perfused with phosphate-buffered saline, followed by perfusion with 20% neutralized formalin. Brains were coronally dissected, fixed overnight at 4 °C, and cryoprotected by means of immersion in a graded sucrose series. Next, the brains were embedded in the OCT compound (Sakura, Tokyo, Japan), and 20 µm thick sections were prepared using a cryostat (CM3050S; Leica Microsystems, Welzer, Germany). The sections were incubated with 25 mM glycine in phosphate-buffered saline for 30 min and incubated with a blocking buffer (5% fetal bovine serum and 0.1% Triton X-100) overnight at 4 °C. Next, the sections were incubated with specific patient or control serum samples diluted at 1:250 dilution in blocking buffer and an anti-ZSCAN1 antibody (catalog no: HPA007938; Sigma Aldrich, St. Louis, MO, USA) for 2 days at 4 °C. Antibody binding to the sections was determined by overnight incubation with the Alexa Fluor 647-conjugated goat anti-human IgG (H+L) cross-adsorbed secondary antibody (1:500 dilution) and an Alexa Fluor 555-conjugated goat anti-rabbit IgG (H+L) secondary antibody (1:500 dilution; catalog no: A21428; ThermoFisher Scientific). The sections were visualized using confocal fluorescence microscopy (FV-1000D; Olympus, Tokyo, Japan).

### 4.4. Detection of Anti-ZSCAN1 and Anti-Na_x_ Autoantibodies by ELISA

Anti-ZSCAN1 and anti-Na_x_ autoantibodies were detected using custom-made ELISAs developed for the detection of anti-ZSCAN1 and anti-Na_x_ autoantibodies, as previously reported [12]. Briefly, the protein synthesis with a GST tag added onto their N-terminus was performed using the wheat germ cell-free translation system. ZSCAN1 protein (Q8NBB4; 380 ng/well) and SCN7A protein fragment containing the major epitope for the anti-Na_x_ antibody (Q01118 amino acid positions 1078–1128; 90 ng/well) were applied to GSH-coated microtiter plates. The negative control protein was prepared using distilled water instead of mRNA during protein preparation. The positive control protein was prepared using mRNA encoding human IgG. The prepared ELISA plates were stored at −80 °C until use and thawed at room temperature before the assay. The wells were incubated with serum samples diluted at 1:100 dilution for 1 h at room temperature. Next, the wells were incubated with a horseradish peroxidase-conjugated goat anti-human IgG (H+L) cross-adsorbed secondary antibody (1:10,000 dilution; catalog no:A18811; Thermo Fisher Scientific) for 1 h at room temperature. Then, the wells were washed three times with TBS-T, and incubated with 100 µL of tetramethylbenzidine–H_2_O_2_ (Pierce TMB substrate kit). Finally, after 30 min of incubation at room temperature, the reaction was stopped by the addition of 100 µL of 2 M H_2_SO_4_ and absorbance was measured at 450 nm. The index value was calculated as follows: [ZSCAN1 or SCN7A absorbance- negative control absorbance]/[positive control absorbance-negative control absorbance] × 100. The cut-off value of each autoantigen was set as follows, based on the mean and SD of negative serum samples: positive (>40 for ZSCAN1; >10 for SCN7A).

## 5. Conclusions

In conclusion, we used the sera of patients with ROHHAD syndrome not associated with a tumor and identified ZSCAN1 as a target antigen for autoantibody response. Following our previous findings showing ZSCAN1 expression in the SFO, we here demonstrated that ZSCAN1 was co-expressed at the site of antibody reactivity to the IgG in the patient serum. Furthermore, 12 of the 14 patients with ROHHAD syndrome not associated with a tumor were found to be positive for anti-ZSCAN1 autoantibodies via an ELISA. Altogether, these results suggest that anti-ZSCAN1 autoantibodies are a feasible diagnostic marker for ROHHAD syndrome not associated with a tumor.

## Figures and Tables

**Figure 1 ijms-25-01794-f001:**
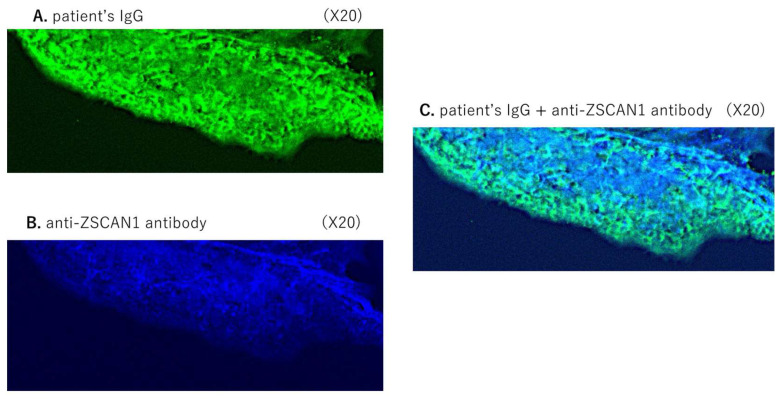
Co-expression of patient serum IgG and anti-ZSCAN1 in the mouse SFO. Immunostaining of mouse SFOs with the serum sample of a patient with ROHHAD syndrome (Case 1). (**A**): Green liner signal indicates IgG deposits (magnification, 20×). (**B**): Blue signals indicate the anti-ZSCAN1deposits. (**C**): Coimmunostaining of mouse SFOs with the same patient’s serum and anti-ZSCAN1 antibody (magnification, 20×). Light blue signals indicate the deposits of the patient’s IgG and the anti-ZSCAN1 antibody.

**Figure 2 ijms-25-01794-f002:**
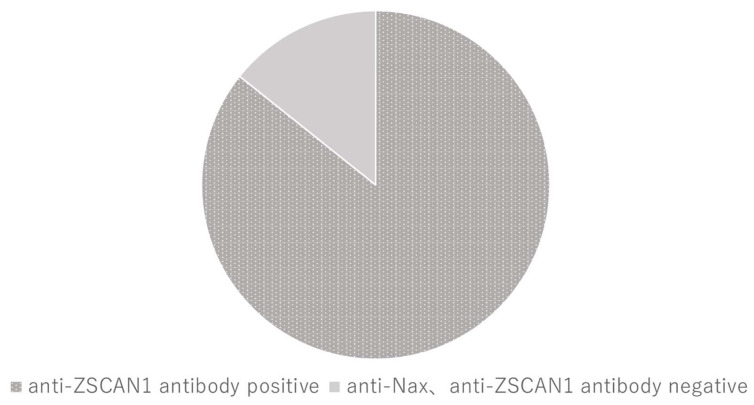
Percentage of ROHHAD patients with anti-ZSCAN1 autoantibodies. ELISA showed that 85.7% (12/14) of the patients with ROHHAD syndrome were positive for anti-ZSCAN1 autoantibodies. Dark and light gray indicate patients who are positive and negative for anti-ZSCAN1 autoantibodies, respectively.

**Table 1 ijms-25-01794-t001:** Clinical characteristics of the patients included in the comprehensive protein array analysis.

	Case 1	Case 2	Case 3	Case 4	Case 5
Age/sex	12/F	14/F	4/F	4/F	5y/M
Serum Na (mEq/L) (normal range:138–145)	153	146	156	149	152
Tumor	-	-	-	+	-
Obesity	+	+	+	+	-
Growth Hormone deficiency	+	+	-	+	-
Anti-SFO autoantibody response	+	+	+	+	+
Anti-Na_x_ autoantibody response	-	-	-	-	-

**Table 2 ijms-25-01794-t002:** Antibody reactivity status of Cases 1–5 determined using a customized panel of 33 candidate proteins initially identified in the comprehensive protein array analysis.

No.	Gene Symbol	Clone ID	①	②	③	④	⑤
1	GPRC5A	FLJ10899AAAF	0.9	18.7	0.2	0.2	0.1
2	HDGFL1	FLJ32262AAAF	3.2	0	−0.5	−0.5	−0.6
3	TRIM34	FLJ14970AAAF	0.3	0.2	−0.2	0	−0.2
4	FICD	FLJ20007AAF	17.4	0.4	0	0	−0.2
5	SMOC2	FLJ32138AAAF	1	0.7	0.1	10.2	0
6	ZSCAN1	FLJ33779AAAF	11.6	72.1	96.2	6.6	189.6
7	ZNF329	FLJ39223AAAF	1.3	3.4	1.7	0.6	47.6
8	REM2	FLJ38964AAAF	0.8	0.4	4	0.2	0.6
9	SREBF1	FLJ33812AAAF	0.8	9.9	0.7	0.9	0
10	GATAD2A	FLJ43986AAAF	3.7	27.4	1.3	1.3	1.3
11	ATG16L1	FLJ4948AAAF	1.3	0.7	0.1	7.9	0.6
12	ZNF641	FLJ31295AAAF	0.7	0.5	−0.1	34	0.2
13	ZNF133	FLJ16801AAAF	0.6	0.4	1.2	54.6	0.2
14	CCDC136	FLJ42456AAAF	1.1	0.2	−0.1	16.5	−0.2
15	ZNF414	FLJ23611AAAF	0.6	9.6	0.4	0.2	0.3
16	EMCN	FLJ94060AAAF	0	0.7	16.7	0	−0.2
17	ZNRD1	FLJ81067AAAF	0.2	0.4	1.2	0.2	0.3
18	HDAC3	FLJ80944AAAF	0.4	0.4	0.2	0	−0.4
19	ADAD2	FLJ96146AAAF	1	16.4	2.7	0.6	0.5
20	APBB1	FLJ54642AAAF	9	0.2	4.5	0.1	−0.2
21	MOK	FLJ93194AAAF	0.7	0.5	0.5	30.6	0.2
22	SHKBP1	FLJ93157AAAF	0.4	0.2	0.2	0.2	0.1
23	PTK2	FLJ57218AAAF	41.2	0.3	−0.4	0.3	−0.4
24	IRX2	FLJ93157AAAF	0.2	0.5	0	53.7	−0.1
25	COL26A1	FLJ57218AAAF	21.2	0.2	0	0	−0.2
26	PPARGC1A	FLJ82376AAAF	0.1	1	1.9	0.3	0
27	IFT74	FLJ13645AAAN	0.7	0.1	4.8	0.1	0.3
28	ATF7IP	FLJ21497AAAN	1.8	0.1	−0.2	−0.2	−0.2
29	TNKS1BP1	FLJ42913AAAN	0.4	0.3	6.2	−0.2	5.5
30	HES1	FLJ20408AAAN	8	0.5	0.1	0.1	0
31	ZC3H18	FLJ84571AAAN	0	22.7	−0.2	0	−0.5
32	AKAP4	N0751-C01-7_FOD_00129	9.8	0.3	0.4	0	−0.2
33	LIPE	N0831-D05-1_FOD_21144	0.2	13.3	−0.1	0	0.1

Light gray indicates weak positivity, and dark gray indicates strong positivity.

**Table 3 ijms-25-01794-t003:** Clinical characteristics of patients diagnosed with ROHHAD syndrome not associated with a tumor (upper) and control patients (lower).

ROHHAD Case	Age/Sex	Serum Na (mEq/L)	Obesity	GHD	Central Apnea	Anti-Nax Antibody	Prognosis	Anti-ZSCAN1 Titer
1	3/F	153	+	+	-	-	Alive	54
2	4/F	157	+	-	-	-	Alive	70
3	3/M	157	+	+	-	-	Alive	70
4	13/M	173	-	-	-	N/A	Alive	59
5	9/M	164	+	+	+	+	Alive	62
6	2/F	147	+	-	+	-	Alive	160
7	17/F	194	+	+	+	-	Death	244
8	10/M	150	+	+	-	-	Alive	0
9	11/M	155	+	+	-	-	Alive	62
10	2/F	143	+	N/A	+	-	Alive	49
11	7/F	150	+	+	-	-	Alive	317
12	12/M	151	+	-	+	-	Alive	105
13	8/M	145	+	-	-	-	Alive	0
14	2/F	167	+	-	+	-	Alive	128
**Control case**	**Age/sex**	**Serum Na (mEq/L)**	**Obesity**	**GHD**	**Central apnea**	**Anti-Nax Antibody**	**Diagnosis**	**Anti-ZSCAN1 titer**
1	12/F	138	-	-	-	-	healthy	0.1
2	3/M	140	-	-	-	-	healthy	2.6
3	1/F	143	-	-	-	-	healthy	1.1
4	2/F	137	-	-	-	-	West syndrome	3.3
5	9/M	139	-	-	-	-	healthy	2.3
6	14/F	140	-	-	-	-	Basedow’s disease	7.8
7	1/M	137	-	-	-	-	healthy	4.3
8	2/M	140	-	-	-	-	healthy	4.8
9	8/M	139	+	-	-	-	Congenital hypothyroidism	4.2
10	13/M	140	-	-	-	-	healthy	0.7
11	15/M	139	-	-	-	-	IDDM	3.2
12	14/F	138	-	-	-	-	Myasthenia gravis	1.2
13	1/M	140	-	-	-	-	healthy	2.5
14	12/F	139	-	-	-	-	Basedow’s disease	7.4
15	13/M	140	-	-	-	-	healthy	2.2

## Data Availability

The data that support the findings of this study are available from the corresponding author upon reasonable request.

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
