# Peer review of "Anti-ZSCAN1 Autoantibodies Are a Feasible Diagnostic Marker for ROHHAD Syndrome Not Associated with a Tumor"

_ijms, 2024, doi:10.3390/ijms25031794_

Round 1
Reviewer 1 Report
Comments and Suggestions for Authors
The authors present a method for diagnosing an even rarer subset of a very rare disease.
Protein array analysis identified ZSCAN1 as an autoantigen in patients with ROHHAD syndrome without a tumor. They used a quantitative, custom-made ELISA to find that more than 85% of ROHHAD syndrome patients who did not have NET had anti-ZSCAN1 autoantibody positivity. Both the method and the identified autoantibody may play an important role in the diagnosis of this rare disease subgroup.
The methods used and the number of patients included are all appropriate.
The figures and tables are clear and illustrative.
The content of the article is generally acceptable.
However, the English language requires major revision, both syntactically and grammatically.
The text contains many mispronunciations, grammatical errors, and poorly worded sentences. A thorough English correction is needed.
Author Response
Response to reviewer 1 comments
Authors’ reply: We thank the reviewer for the careful reading of our paper and for the unbiased comments. We appreciate the reviewer’s comments. There has been a continuous effort to identify the exact cause of ROHHAD syndrome since it was initially proposed. The titer of anti-ZSCAN1 antibodies, which have been proposed as a new diagnostic marker, point towards the autoimmune hypothesis, although this has to be judged by the reader in the light of other reports of ROHHAD syndrome.
Reviewer 2 Report
Comments and Suggestions for Authors
The manuscript reported that ZSCAN1 can be a diagnostic marker for ROHHAD syndrome patients without tumors. The authors first identified detectable ZSCAN1 protein in 4 ROHHAD patients without tumors and not detectable in 1 patient with a tumor. The authors also validated using ZSCAN1 as the diagnostic biomarker in serum samples from 14 ROHHAD patients without tumors. 12/14 patients were tested positive for ZSCAN1 detection. The study is interesting and needed given the current incurable situation of ROHHAD patients. However, key data is missing to support the conclusion:
-
Validation of ZSCAN1 staining in mouse SFO tissue specimens is not well conducted. The staining quality is overall bad. The so-claimed signal could be an unspecific background noise. Please perform the staining with proper co-staining, including a negative IgG control and DAPI nuclei staining. No conclusion can be drawn from the current figure.
-
Interestingly, ZSCAN1 was only detectable in ROHHAD patients without tumors. As stated in the introduction, the ZSCAN1 was first detected in ROHHAD patients with NETs. Please discuss why ZSCAN1 was not detectable in your samples (depending on tumor type?)
-
The Table 2 and Figure 2 only presented a part of the data. Please include clinic data and the level of ELISA of ZSCAN1 in healthy patients and patients with tumors.
Minor English editing is needed.
Author Response
Response to reviewer 2 comments
The manuscript reported that ZSCAN1 can be a diagnostic marker for ROHHAD syndrome patients without tumors. The authors first identified detectable ZSCAN1 protein in 4 ROHHAD patients without tumors and not detectable in 1 patient with a tumor. The authors also validated using ZSCAN1 as the diagnostic biomarker in serum samples from 14 ROHHAD patients without tumors. 12/14 patients were tested positive for ZSCAN1 detection. The study is interesting and needed given the current incurable situation of ROHHAD patients. However, key data is missing to support the conclusion:
Author’s reply:
We thank you for your careful and suggestive comments. We revised the paper based on the reviewer 2 comments. We described the each response point by point.
1. Validation of ZSCAN1 staining in mouse SFO tissue specimens is not well conducted. The staining quality is overall bad. The so-claimed signal could be an unspecific background noise. Please perform the staining with proper co-staining, including a negative IgG control and DAPI nuclei staining. No conclusion can be drawn from the current figure.
Author’s reply: We thank you about the correct comments. We could not upload the precise figure1-c. That’s why we revised and submitted again the correct figure 1-C.
2. Interestingly, ZSCAN1 was only detectable in ROHHAD patients without tumors. As stated in the introduction, the ZSCAN1 was first detected in ROHHAD patients with NETs. Please discuss why ZSCAN1 was not detectable in your samples (depending on tumor type?)
Author’s reply: We thank you the appropriate suggestion. We also considered the case not associated tumor was not detected anti- ZSCAN1 antibody. In the case, she was associated with lymphangioma. Typical ROHHAD-NET was associated with tumors such as neuroganglioma. The difference of tumor types has an influence of not existence of anti-ZSCAN1 antigen. We added about that point in the discussion.
3. The Table 2 and Figure 2 only presented a part of the data. Please include clinic data and the level of ELISA of ZSCAN1 in healthy patients and patients with tumors.
Author’s reply: We added the table 2 indicating clinical data of the control subjects. All of control subjects exhibited anti-ZSCAN1 antibody index were <40.
Round 2
Reviewer 2 Report
Comments and Suggestions for Authors
I appreciate all the edits and changes the authors made to address my concerns. However, there are still some major concerns related with the modified manuscript:
1. I understand the limits of uploading images with high resolution. However, the staining on the Figure 1 is not publishable. Cell shape and nuclei shape were not distinguishable. The staining looks like an autofluorescent background. The manuscript looks very similar to the published paper: https://onlinelibrary.wiley.com/doi/10.1002/ana.26380. Please refer to the staining published there.
2. I don't think putting <40 in all the control samples is a good practice for the science rigor. Please update the detected data and score in the table.
Comments on the Quality of English LanguageMinor language edit is needed
Author Response
I appreciate all the edits and changes the authors made to address my concerns. However, there are still some major concerns related with the modified manuscript:
Author’s reply:
We thank you for your careful and suggestive comments. We revised the paper based on the reviewer 2 comments. We described each response point by point as the following. Also, we revised the paper according to the MDPI English editing
1. I understand the limits of uploading images with high resolution. However, the staining on the Figure 1 is not publishable. Cell shape and nuclei shape were not distinguishable. The staining looks like an autofluorescent background. The manuscript looks very similar to the published paper: https://onlinelibrary.wiley.com/doi/10.1002/ana.26380. Please refer to the staining published there.
Author’s reply: We thank you about the precise comments. We revised again Fig1.C. We have already performed the immunostaining in mouse subfornical organ as same methods as previously published in the several journal. (For example: https://www.ncbi.nlm.nih.gov/pmc/articles/PMC8029419/ (Fig.2)). Therefore, unfortunately we didn’t perform immunostaining by DAPI. In addition, we recognized the differences of tissue from subfornical organ in our experiment and tumor tissue in the reviewer’s indicated experiment. This result offers the new insight that the anti-ZSCAN1 antibody existed in the subfornical organ and have an influence of hypothalamic dysfunction such as adipsic hypernatremia.
2. I don't think putting <40 in all the control samples is a good practice for the science rigor. Please update the detected data and score in the table.
Author’s reply: We thank you about the correct comments. We added each titer level of control subjects in Table 2.

Round 3
Reviewer 2 Report
Comments and Suggestions for Authors
The authors have made efforts to respond to my suggestions. The staining figure looks better now with a clear signal. I have no other suggestions.